# Virtual Water Flows Embodied in International and Interprovincial Trade of Yellow River Basin: A Multiregional Input-Output Analysis

**Guiliang Tian [1], Xiaosheng Han [1], Chen Zhang [2,\*], Jiaojiao Li [1] and Jining Liu [1]**

[1] Business School, Hohai University, Nanjing 211100, China; tianguiliang@hhu.edu.cn (G.T.); hxs7758@hhu.edu.cn (X.H.); lijiaojiao@hhu.edu.cn (J.L.); liujining@hhu.edu.cn (J.L.)

[2] Department of Information Systems and Analytics, Bryant University, Smithfield, RI 02917, USA

[\*] Correspondence: czhang@bryant.edu

**Abstract:** With the imminent need of regional environmental protection and sustainable economic development, the concept of virtual water is widely used to solve the problem of regional water shortage. In this paper, nine provinces, namely Qinghai, Sichuan, Gansu, Ningxia, Inner Mongolia, Shaanxi, Shanxi, Henan, and Shandong in the Yellow River Basin (YRB), are taken as the research objects. Through the analysis of input-output tables of 30 provinces in China in 2012, the characteristics of virtual water trade in this region are estimated by using a multi-regional input-output (MRIO) model. The results show that: (1) The YRB had a net inflow of 17.387 billion m$^3$ of virtual water in 2012. In interprovincial trade, other provinces outside the basin export 21.721 billion m$^3$ of virtual water into the basin. In international trade, the basin exports 4334 million m$^3$ of virtual water to the international market. (2) There are different virtual flow paths in the basin. Shanxi net inputs virtual water by interprovincial trade and international trade, while Gansu and Ningxia net output virtual water by interprovincial trade and international trade. The other six provinces all net output virtual water through international trade, and obtain the net input of virtual water from other provinces outside the basin. (3) From the industrial structure of the provinces in the basin, the provinces with a relatively developed economy, such as Shandong and Shanxi, mostly import virtual water in the agricultural sector, while relatively developing provinces, such as Gansu and Ningxia, mostly import virtual water in the industrial sector. In order to sustain the overall high-quality development of the YRB, we propose the virtual water trade method to quantify the net flow of virtual water in each province and suggest the compensation responsibility of the virtual water net inflow area, and the compensation need of the virtual water net outflow area, in order to achieve efficient water resources utilization.

**Keywords:** virtual water flows; multiregional input-output model; Yellow River Basin; international and interprovincial trade

## 1. Introduction

Rapid population growth and economic development have exacerbated energy and water consumption [1]. With the sharp increase of these natural resources consumption, the scarcity becomes a bottleneck for the sustainable social and economic development of the region [2]. The water resources shortage now hurdles the continuous economic development of many regions [3,4]. Over the past six decades, with the increase in population and socio-economic development, the YRB's water consumption (including agriculture, industrial, and domestic water) and the number of water facilities have increased significantly [5,6]. Although the economic development of the YRB is remarkable,

the ecological environment of the basin is extremely fragile, and the water resource guarantee situation is grim. Therefore, improving the management to allow better value-adding with limited water resources is an important step in realizing the national strategy of ecological protection and high-quality development in the YRB [7].

The Yellow River, as the second-longest river in China, originates from the Qinghai-Tibet Plateau and flows through Qinghai, Sichuan, Gansu, Ningxia, Inner Mongolia, Shaanxi, Shanxi, Henan, and Shandong Province [8]. As an essential water supply source in northwest and north China, the Yellow River has an annual average runoff of 53.48 billion m$^3$, accounting for only about 2% of the country's river runoff. The per capita annual runoff is 473 m$^3$, which is only 23% of the national per capita annual runoff. However, it is responsible for 15% of farmland water and 12% of the population's water supply and for diverting water to Tianjin, Qingdao, and other areas [9]. Due to the competitive utilization of water resources in nine provinces of the YRB, the Yellow River has been cut off (i.e., the water flow of the Yellow River was too low) since the 1970s. In the 27 years from 1972 to 1998, the Yellow River was cut off for 22 years, which had a serious negative impact on the regional climate. After 1999, the Yellow River conservancy commission took strict administrative measures to realize the continuous flow of the Yellow River during the years of great drought, but given that more than 20 billion cubic meters of water are needed annually to transport sediment into the sea to slow down silting in the lower reaches of the Yellow River, the continuous flow is temporary, not ecologically sustainable.

As shown in Figure 1, North-China is a region with water shortage. According to the international standard, water shortage is moderate if the per capita water resource is less than 2000 m$^3$, severe if the per capita water resource is less than 1000 m$^3$, and extreme if the per capita water resource is less than 500 m$^3$. Except for Qinghai and Sichuan, the other seven provinces are in a state of water shortage. Specifically, Shaanxi and Inner Mongolia have moderate water shortage; Gansu has severe water shortage; Ningxia, Shanxi, Shandong and Henan have extreme water shortage [10]. After many scholars' research on the Yellow River Basin, main reasons for the shortage of water resources are as follows: Firstly, the soil erosion in the Loess Plateau is severe [11,12]. To control soil erosion, the Chinese government has implemented a series of large-scale ecological projects [11], while reducing the runoff of the Yellow River [13,14]. Secondly, weather factors, such as rising temperatures, reduced precipitation, and extreme weather events, have also exacerbated the water shortage in the Yellow River [15]. At the end of the 20th century, the Yellow River was almost completely dry [16]. Thirdly, the economic development consumed a lot of water. The "Western Development" strategy has enabled the central and western provinces to obtain economic benefits [17], but it has exacerbated local water consumption and caused severe water pollution. Finally, the agricultural irrigation consumed a lot of water [3].

In order to solve the problem of water shortage in northern China, the Chinese government implemented the South-to-North Water Diversion Project. To a certain extent, this artificial water channel from the water-rich areas to the water-poor areas can alleviate the water pressure in the north, but it has potentially disturbed the water cycle balance in the water sourcing area. Meanwhile, the virtual water strategy provides a new way to alleviate the pressure of regional water shortage. Based on the perspective of consumption, it transfers water-intensive products through trading to solve the problem of unbalanced distribution of regional water resources [17].

Because of the shortage of water resources in the YRB, it is very important to save water resources. Therefore, We must commercialize water resources and optimize the allocation of water resources to the most needed uses. We need to analyze the amount of water resources used by various industrial sectors in the field of production, as well as the inflow and outflow of water resources in the YRB along with the trade of commodities. Virtual water theory and method is an effective method for water consumption analysis. Therefore, this study measures and analyzes the virtual water quantity, inflow and outflow of multi regions in the YRB, so as to discuss how to optimize the allocation of water resources in the YRB. Virtual water was first proposed by Allan [18], which refers to the amount of water resources needed in the production of products and services [19]. It is a concept closely linked with "water footprint," defined as the amount of water resources required for all products and services

consumed by a person (a country, a region) in a certain period of time [20]. Methods for calculating regional water footprints can be generally classified into two groups, namely the bottom-up method and the input-output analysis (IOA) method [21]. The bottom-up method was first introduced by Hoekstra and Chapagain, which is the sum of all goods and services consumed multiplying with their corresponding virtual water contents (VWC) [22]. However, this method can only be applied to agricultural products and crops, but not to industrial and tertiary products. Because this method ignores the dependence between economic sectors, it is difficult to track the entire industrial supply chain [23].

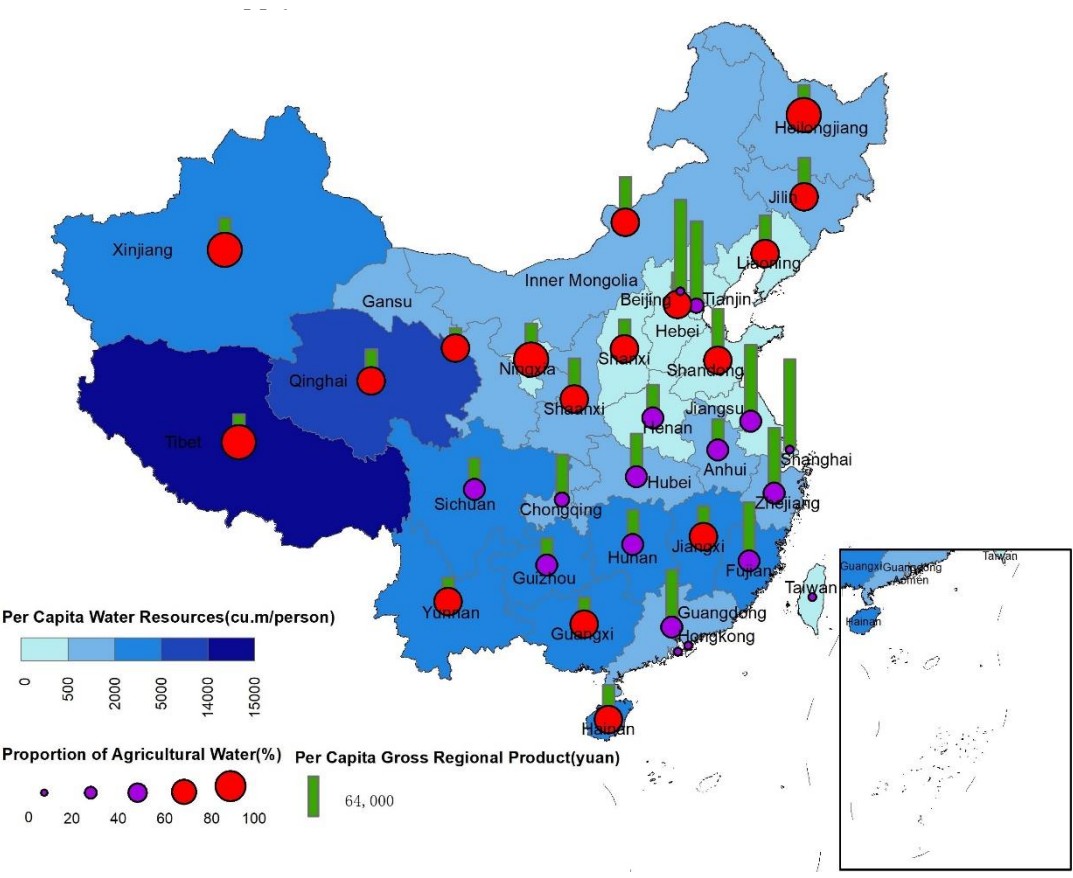

**Figure 1.** Differences in per capita water resources, the proportion of agricultural water, and per capita gross regional product of China in 2017. Data sources: China Statistical Yearbook 2018. Hongkong, Macau, and Taiwan are excluded due to data availability.

The input-output analysis (IOA) method is a top-down method that uses sectoral monetary transaction data to account for the complex interdependencies of industrial sectors [24,25]. The IOA method is helpful to distinguish direct virtual water consumption from indirect virtual water consumption and consider the intermediate input of products to avoid double-count [26]. And the IOA method calculates the direct and indirect water consumption in the final consumption by establishing the extended water input-output table, taking the water utilization as the final consumption [27,28]. Moreover, the IOA method is more suitable for revealing virtual water flows embodied in trade, and it has the advantages of simplicity and relatively good data availability [23].

The IOA method can be further classified into the single-regional-input-output model (SRIO) and the multi-regional input-output model (MRIO). The SRIO method assumes that imported goods and services are of the same technology as those produced in the same domestic sector. In addition, SRIO was unable to calculate virtual water flows between multiple areas with trade links. While the MRIO method overcomes the above shortcomings of SRIO by linking all trade between sectors in each

region [22]. Therefore, MRIO is commonly used in the study of virtual water trade among different regions [29–31]. For example, Zhang applied the multi-region input-output table of 33 sectors and 33 regions in 2002, to calculate the virtual water flow in Beijing [32]. Feng constructed a multi-regional input-output table of 48 sectors and 4 regions to study the consumption-driven virtual water trade in the YRB [3].

The existing literature on virtual water includes both global level and that of a specific country or region, especially provinces with water shortage. At the global level study, Arto (2016)and Chen (2013)investigated water use, water footprint, and water trade balance among global countries using the MRIO method [33,34]. In comparison with the global level, more virtual water studies focus on a particular water-scare region. For example, Zhang (2011) investigated the water footprint of a scarce water city, Beijing, and found that Beijing was a virtual water importer [32], while Dong (2013)studied the water footprint of a water-scarce province, Liaoning, and found that Liaoning was a virtual water exporter, which aggravated the local water shortage situation [21]. Zhang and Anadon (2014) also found that China's virtual water trade flows from the water-scare areas in the north to the water-rich areas in the south. Hence the virtual water outflow further aggravates the water shortage situation in the north [35].

There are significant differences in water endowments between the nine YRB provinces. Qinghai and Sichuan are relatively abundant in water resources, while the other provinces have a severe shortage of water resources. Besides, due to the frequent occurrence of extreme weather, the shortage of water resources in the Yellow River Basin has further intensified. Studying the virtual water trade of the Yellow River is of considerable significance to solve the problem of water shortage in the middle and lower reaches of the Yellow River while ensuring the ecological protection and sustainable development of the upstream water source provinces and realizing the overall high-quality development of the YRB. However, the existing studies usually focus on a certain administrative region, such as a province, but very few studied the basin scale with the overall characteristics of the whole water system, such as the YRB. Only Feng (2012) have established a multi-regional input-output table of 48 sectors and four regions to calculate the virtual water volume of the YRB [3]. Therefore, this paper selects nine provinces of the YRB as the research object, analyzes the overall and industrial characteristics of the virtual flow during the internal and external trading. By revealing differences of the virtual flow pattern in each province of the YRB, we provide a theoretical reference and decision-making basis for the high-quality development of the YRB.

## 2. Materials and Method

### 2.1. Single-Region Input-Output Model

Input-output analysis, first proposed by economist [36], can reflect the direct and indirect links between the production activities of various industrial sectors in the national economic system, so it is widely used to measure the virtual water in trade. According to the input-output analysis, the production activities of a complete economic system have the following balance:

$$
\begin{aligned}
x_1 &= z_{11} + z_{12} + \cdots + z_{1n} + f_1 \\
x_2 &= z_{21} + z_{22} + \cdots + z_{2n} + f_2 \\
&\quad\vdots \\
x_n &= z_{n1} + z_{n2} + \cdots + z_{nn} + f_n
\end{aligned}
\tag{1}
$$

Equation (1) can be organized into the following form:

$$
x_i = \sum_{j=1}^{n} z_{ij} + f_i
\tag{2}
$$

where, $x_i$ is the total output of sector $i$, $z_{ij}$ is the intermediate input from sector $i$ to sector $j$, $f_i$ is the final consumption of sector $i$.

The technical coefficient $a_{ij} = \frac{z_{ij}}{x_j}$, means that the input from sector $i$ is transformed into the output of sector $j$ by the production technologies. Therefore, Equation (2) can be rewritten as:

$$x_i = \sum_{j=1}^{n} a_{ij} x_j + f_i \tag{3}$$

### 2.2. Multi-Regional Input-Output Model of the YRB

The single-region input-output model can only reveal the inflow and outflow of virtual water in a single province [22], while the Yellow River flows through the nine provinces. Studying the YRB requires consideration of the relationship between multiple provinces with trade. Therefore, it is necessary to break through the limitations of the single-region input-output model and adopt a multi-regional input-output (MRIO) accounting framework. According to the regional characteristics and industrial conditions of the YRB, Table 1 constructs a multi-regional input-output table for 30 provinces/autonomous regions/municipalities (except Tibet Autonomous Region, Taiwan Region, Hong Kong, and Macao Special Administrative Region) with 30 industrial sectors. The table contains ten regions (9 provinces in the YRB as nine regions, and the remaining 21 provinces outside the YRB are classified as one region). Based on the principle of the single-region input-output model, according to the multi-regional input-output table of the YRB, the intermediate use of sector 1 in region 1 is allocated to other sectors in region 1 and other sectors in other regions, shown in Equation (4):

$$
\begin{aligned}
x_1^{11} &= a_{11}^{11} x_1^1 + a_{12}^{11} x_1^1 + \cdots + a_{1,30}^{11} x_1^1 = \sum_{j=1}^{30} a_{1j}^{11} x_1^1 \\
x_1^{12} &= a_{11}^{12} x_1^1 + a_{12}^{12} x_1^1 + \cdots + a_{1,30}^{12} x_1^1 = \sum_{j=1}^{30} a_{1j}^{12} x_1^1 \\
&\quad\vdots \\
x_1^{1,10} &= a_{1,1}^{1,10} x_1^1 + a_{1,2}^{1,10} x_1^1 + \cdots + a_{1,30}^{1,10} x_1^1 = \sum_{j=1}^{30} a_{1,j}^{1,10} x_1^1
\end{aligned}
\tag{4}
$$

On the basis of Equation (4), the total output of sector 1 in region 1 is calculated as follows:

$$
\begin{aligned}
x_1^1 &= x_1^{11} + x_1^{12} + \cdots + x_1^{1,10} + f_1^1 + e_1^1 \\
&= \sum_{j=1}^{30} a_{1j}^{11} x_1^1 + \sum_{j=1}^{30} a_{1j}^{12} x_1^1 + \cdots + \sum_{j=1}^{30} a_{1j}^{1,10} x_1^1 + f_1^1 + e_1^1 \\
&= \sum_{s=1}^{10} \sum_{j=1}^{30} a_{1j}^{1s} x_1^1 + f_1^1 + e_1^1
\end{aligned}
\tag{5}
$$

On the basis of Equation (5), the total output of region 1 is calculated as follows:

$$
\begin{aligned}
x^1 &= x_1^1 + x_2^1 + \cdots + x_{30}^1 \\
&= \sum_{s=1}^{10} \sum_{j=1}^{30} a_{1j}^{1s} x_1^1 + f_1^1 + e_1^1 + \sum_{s=1}^{10} \sum_{j=1}^{30} a_{2j}^{1s} x_1^1 + f_2^1 + e_2^1 \\
&\quad + \cdots + \sum_{s=1}^{10} \sum_{j=1}^{30} a_{30,j}^{1s} x_1^1 + f_{30}^1 + e_{30}^1 \\
&= \sum_{i=1}^{30} \sum_{s=1}^{10} \sum_{j=1}^{30} a_{ij}^{1s} x_i^1 + \sum_{i=1}^{30} f_i^1 + \sum_{i=1}^{30} e_i^1
\end{aligned}
\tag{6}
$$

**Table 1.** Multi-regional input-output table of the Yellow River Basin (YRB).

| | | | Intermediate Use | | | | Final Use | | | | Export | Total Output |
|---|---|---|---|---|---|---|---|---|---|---|---|---|
| | | | Qinghai | ··· | Shandong | Other Provinces | Qinghai | ··· | Shandong | Other Provinces | | |
| | | | Sector1···Sector30 | ··· | Sector1···Sector30 | Sector1···Sector30 | | | | | | |
| **Intermediate 5** | **Qinghai** | **Sector 1** | $z_{1,1}^{1,1}$ ··· $z_{1,30}^{1,1}$ | ··· | $z_{1,1}^{1,9}$ ··· $z_{1,30}^{1,9}$ | $z_{1,1}^{1,10}$ ··· $z_{1,30}^{1,10}$ | $f_1^{1,1}$ ··· | $f_1^{1,9}$ | $f_1^{1,10}$ | | $e_1^1$ | $Z_1^1$ |
| | | ··· | ··· ··· ··· | ··· | ··· ··· ··· | ··· ··· ··· | ··· ··· | ··· | ··· | | ··· | ··· |
| | | **Sector 30** | $z_{30,1}^{1,1}$ ··· $z_{30,30}^{1,1}$ | ··· | $z_{30,1}^{1,9}$ ··· $z_{30,30}^{1,9}$ | $z_{30,1}^{1,10}$ ··· $z_{30,30}^{1,10}$ | $f_{30}^{1,1}$ ··· | $f_{30}^{1,9}$ | $f_{30}^{1,10}$ | | $e_{30}^1$ | $Z_{30}^1$ |
| | ··· | ··· | ··· ··· ··· | ··· | ··· ··· ··· | ··· ··· ··· | ··· ··· | ··· | ··· | | ··· | ··· |
| | **Shandong** | **Sector 1** | $z_{1,1}^{9,1}$ ··· $z_{1,30}^{9,1}$ | ··· | $z_{1,1}^{9,9}$ ··· $z_{1,30}^{9,9}$ | $z_{1,1}^{9,10}$ ··· $z_{1,30}^{9,10}$ | $f_1^{9,1}$ ··· | $f_1^{9,9}$ | $f_1^{9,10}$ | | $e_1^9$ | $Z_1^9$ |
| | | ··· | ··· ··· ··· | ··· | ··· ··· ··· | ··· ··· ··· | ··· ··· | ··· | ··· | | ··· | ··· |
| | | **Sector 30** | $z_{30,1}^{9,1}$ ··· $z_{30,30}^{9,1}$ | ··· | $z_{30,1}^{9,9}$ ··· $z_{30,30}^{9,9}$ | $z_{30,1}^{9,10}$ ··· $z_{30,30}^{9,10}$ | $f_{30}^{9,1}$ ··· | $f_{30}^{9,9}$ | $f_{30}^{9,10}$ | | $e_{30}^9$ | $Z_{30}^9$ |
| | **Other Provinces** | **Sector 1** | $z_{1,1}^{10,1}$ ··· $z_{1,30}^{10,1}$ | ··· | $z_{1,1}^{10,9}$ ··· $z_{1,30}^{10,9}$ | $z_{1,1}^{10,10}$ ··· $z_{1,30}^{10,10}$ | $f_1^{10,1}$ ··· | $f_1^{10,9}$ | $f_1^{10,10}$ | | $e_1^{10}$ | $Z_1^{10}$ |
| | | ··· | ··· ··· ··· | ··· | ··· ··· ··· | ··· ··· ··· | ··· ··· | ··· | ··· | | ··· | ··· |
| | | **Sector 30** | $z_{30,1}^{10,1}$ ··· $z_{30,30}^{10,1}$ | ··· | $z_{30,1}^{10,9}$ ··· $z_{30,30}^{10,9}$ | $z_{30,1}^{10,10}$ ··· $z_{30,30}^{10,10}$ | $f_{30}^{10,1}$ ··· | $f_{30}^{10,9}$ | $f_{30}^{10,10}$ | | $e_{30}^{10}$ | $Z_{30}^{10}$ |
| **Import** | | | $I_1^1$ ··· $I_{30}^1$ | ··· | $I_1^9$ ··· $I_{30}^9$ | $I_1^{10}$ ··· $I_{30}^{10}$ | | | | | | |
| **Value added** | | | $V_1^1$ ··· $V_{30}^1$ | ··· | $V_1^9$ ··· $V_{30}^9$ | $V_1^{10}$ ··· $V_{30}^{10}$ | | | | | | |
| **Total input** | | | $Z_1^1$ ··· $Z_{30}^1$ | ··· | $Z_1^9$ ··· $Z_{30}^9$ | $Z_1^{10}$ ··· $Z_{30}^{10}$ | | | | | | |

Note: "Other Provinces" is the remaining 21 provinces beside the nine provinces of the Yellow River Basin.

On the basis of Equation (6), the total output of each region is calculated as follows:

$$
\begin{aligned}
X \;=\;& x^1 + x^2 + \cdots + x^{10} \\
=\;& \sum_{i=1}^{30}\sum_{s=1}^{10}\sum_{j=1}^{30} a_{ij}^{1s} x_i^1 + \sum_{i=1}^{30} f_i^1 + \sum_{i=1}^{30} e_i^1 + \sum_{i=1}^{30}\sum_{s=1}^{10}\sum_{j=1}^{30} a_{ij}^{2s} x_i^2 + \sum_{i=1}^{30} f_i^2 + \sum_{i=1}^{30} e_i^2 \\
&+ \cdots + \sum_{i=1}^{30}\sum_{s=1}^{10}\sum_{j=1}^{30} a_{ij}^{10,s} x_i^{10} + \sum_{i=1}^{30} f_i^{10} + \sum_{i=1}^{30} e_i^{10} \\
=\;& \sum_{r=1}^{10}\sum_{i=1}^{30}\sum_{s=1}^{10}\sum_{j=1}^{30} a_{ij}^{rs} x_i^r + \sum_{r=1}^{10}\sum_{i=1}^{30} f_i^r + \sum_{r=1}^{10}\sum_{i=1}^{30} e_i^r
\end{aligned}
\tag{7}
$$

where, $x_i^r$ is the total output of sector $i$ in region $r$; $a_{ij}^{rs}$ is the direct input coefficient; $f_i^r$ is the final demand of sector $i$ in region $r$; $e_i^r$ is the virtual water exported by sector $i$ of region $r$ to the final demand of other regions;

In order to simplify the calculation process, Equation (7) can be expressed in the following matrix form:

$$
X = A^{rs} X + F + E \tag{8}
$$

Transforming the Equation (8), we obtain the input-output model of 10 regions in the YRB:

$$
X = (I - A^{rs})^{-1}(F + E) = L(F + E) \tag{9}
$$

$$
L = (I - A^{rs})^{-1} = 
\begin{bmatrix}
l^{11} & l^{12} & \cdots & l^{1,10} \\
l^{21} & l^{22} & \cdots & l^{2,10} \\
\vdots & \vdots & \ddots & \vdots \\
l^{10,1} & l^{10,2} & \cdots & l^{10,10}
\end{bmatrix}
\tag{10}
$$

where, $X$, $I$, $F$, $E$, $A^{rs}$, respectively, represent the output matrix, the unit matrix, the final demand matrix, the export matrix, and the direct input coefficient matrix; $L = (I - A^{rs})^{-1}$ is the Leontief inverse matrix and the element $l_{ij}^{rs}$ is expressed as the output of sector $i$ in region $r$ satisfying the unit final demand of sector $j$ in region $s$.

In order to build a multi-regional input-output model of the YRB with water resource expansion, it is necessary to introduce the water consumption of each sector based on the original model. The direct water use coefficient $y_i^r$ in the production process of each sector is the basis for virtual water volume accounting which represents the direct water consumption of sector $i$ required to produce a unit product of sector $i$ in region $r$. It is calculated in the form of Equation (11).

$$
y_i^r = \frac{w_i^r}{x_i^r} \tag{11}
$$

where, $w_i^r$ is the direct water consumption required to produce sector $i$ in region $r$; $x_i^r$ is the total output of sector $i$ in region $r$.

The direct water coefficient matrix of the $r$ region is $y^r = \begin{bmatrix} y_1^r & 0 & 0 \\ 0 & \ddots & 0 \\ 0 & 0 & y_{30}^r \end{bmatrix}$, then this direct water coefficient matrix of 10 regions constitutes the complete direct water coefficient matrix, $Y = \begin{bmatrix} y^1 & 0 & 0 \\ 0 & \ddots & 0 \\ 0 & 0 & y^{10} \end{bmatrix}$. $y^r$ is the submatrix of the matrix $Y$. According to the direct water use coefficient, the total water use coefficient can be calculated as follows:

$$Q = YL = Y(I - A^{rs})^{-1} = \begin{bmatrix} q^{11} & q^{12} & \cdots & q^{1,10} \\ q^{21} & q^{22} & \cdots & q^{2,10} \\ \vdots & \vdots & \ddots & \vdots \\ q^{10,1} & q^{10,2} & \cdots & q^{10,10} \end{bmatrix} \tag{12}$$

where, $q^{rs}$ represents the total water consumption (including direct and indirect water consumption) provided by region $r$ satisfying the final demand for one unit of all sectors in region $s$. In particular, this study mainly considers blue water, that is, water resources extracted from rivers, lakes or underground during the production process. Because we mainly consider the optimal allocation of blue water resources, at the same time, the opportunity cost of blue water resources is more obvious, while the use of green water (rainwater retained by the soil as humidity) is relatively singular, for example, it is more suitable for agriculture, research from the perspective of blue water resources is more helpful to promote the better allocation of water resources.

*2.3. Calculation Model of Virtual Water Trade Flow Based on MRIO in the YRB*

Based on the multi-regional input-output model of water resources expansion in the YRB constructed in Section 2.2, the export virtual water volume of each province in the YRB in international trade can be calculated as:

$$VWE = QE \tag{13}$$

We assume that the technical coefficient of other countries outside the system is the same as that of China, so we can calculate the import virtual water volume as follows:

$$VWM = QM \tag{14}$$

where, $M$ represents the value of products imported from other countries, which is obtained by transposing $I$ in the MRIO table.

In domestic interprovincial trade, the calculation formula of virtual water flow in 10 regions of the YRB is shown in Equation (15):

$$VWT^{rs} = \sum_{m=1}^{10} q^{rm} f^{ms} \tag{15}$$

where, $VWT^{rs}$ represents the virtual volume of water transferred from region $r$ to $s$.

Specifically, the virtual flow between the nine provinces of the YRB and other provinces is calculated as follows:

$$VWI = \sum_{s=1}^{9} VWI^s = \sum_{s=1}^{9} \sum_{m=1}^{9} q^{10,m} f^{ms} \tag{16}$$

$$VWO = \sum_{r=1}^{9} VWO^r = \sum_{r=1}^{9} \sum_{m=1}^{9} q^{rm} f^{m,10} \tag{17}$$

where $VWI^s$, is the virtual water input from the external provinces to the provinces of the YRB; $VWO^r$, is the virtual water output from the provinces of the YRB to the external provinces.

*2.4. Data Source*

This paper uses the MRIO table, including China's 30 provinces/autonomous regions/municipalities in 2012 [37]. The table is compiled by researchers from the China Carbon Accounting Database (CEADs) and includes 30 industrial sectors (one agricultural sector, 24 industrial sectors, and five service sectors). The 2012 MRIO table used in this paper is the latest input-output data in China. Because it takes a lot of human, material, and financial resources to compile the input-output table, China compiles the

input-output table every five years, that is, China only publishes the annual input-output data in the year with a tail number of 7 or 2, but usually postpones the publication for three years. Therefore, the 2017 China input-output table has not yet been published, and 2012 is the latest input-output data of China at present. Based on the input-output data, the 2012 MRIO table [38] is the latest data. Also, considering the availability of water data, this paper combines five service departments into one department. The actual water consumption data for agriculture, industry, and services are all derived from the China Statistical Yearbook (2013) [39]. Since there is no detailed official water use data for various industries, this paper obtains water consumption data for various industries through an indirect method. That is, water consumption in the industrial sector is allocated in a certain proportion of the total water consumption in various industrial sectors. Moreover, the percentage is that the industrial sector is divided by the sum of various industrial sectors in the intermediate consumption of the water production/supply industry [40].

## 3. Results

### 3.1. Overview of Virtual Water Trade in the YRB

Virtual water trade in the YRB is mainly divided into three parts: The trade between provinces in the YRB, the trade between provinces in the YRB and other provinces, and the international trade of provinces in the YRB. Among them, the first two parts of the virtual water trade belong to the domestic interprovincial trade. Table 2 lists the virtual water inflow and outflow of each province in the YRB. YRB has an overall virtual water net inflow of 17.387 billion $m^3$ in 2012. The virtual flow momentum of domestic interprovincial trade is far greater than that of international trade. In domestic interprovincial trade, the virtual water outflow of the YRB is 42.503 billion $m^3$, the inflow is 64.224 billion $m^3$, and the net inflow is 21.721 billion $m^3$. While, in international trade, the virtual water outflow of the YRB is 14.468 billion $m^3$, the inflow is 10.134 billion $m^3$, and the net outflow is 4.334 billion $m^3$. Therefore, in the process of economic development in the YRB, virtual water is mainly introduced through domestic interprovincial trade to alleviate the shortage of water resources.

**Table 2.** Virtual water flows and compositions of each province in the YRB (billion $m^3$).

| Region | Virtual Water Inflow | | | | Virtual Water Outflow | | | | Net Inflow |
|---|---|---|---|---|---|---|---|---|---|
| | From YRB | From Other Provinces | Import | Total Inflow | From YRB | From Other Provinces | Export | Total Outflow | |
| Shanxi | 1.146 | 3.932 | 0.336 | 5.414 | 0.211 | 0.607 | 0.273 | 1.091 | 4.322 |
| Inner Mongolia | 1.369 | 7.827 | 2.379 | 11.575 | 2.506 | 6.912 | 2.493 | 11.911 | −0.336 |
| Shandong | 2.776 | 14.004 | 3.587 | 20.367 | 0.841 | 2.493 | 5.332 | 8.666 | 11.701 |
| Henan | 2.132 | 11.463 | 1.364 | 14.959 | 2.801 | 8.399 | 1.986 | 13.186 | 1.773 |
| Sichuan | 0.989 | 4.619 | 0.843 | 6.451 | 1.048 | 3.256 | 1.564 | 5.867 | 0.584 |
| Shaanxi | 1.960 | 6.355 | 0.700 | 9.015 | 1.649 | 3.336 | 1.256 | 6.241 | 2.774 |
| Gansu | 0.808 | 2.596 | 0.644 | 4.048 | 2.030 | 4.223 | 0.699 | 6.952 | −2.904 |
| Qinghai | 0.307 | 0.747 | 0.095 | 1.149 | 0.222 | 0.450 | 0.180 | 0.851 | 0.298 |
| Ningxia | 0.360 | 0.835 | 0.184 | 1.380 | 0.539 | 0.980 | 0.685 | 2.205 | −0.825 |
| YRB | 11.847 | 52.377 | 10.134 | 74.358 | 11.847 | 30.656 | 14.468 | 56.971 | 17.387 |

Details of the virtual water flow in each province of the YRB are shown in Figure 2. When looking at inflows, Shandong Province has the largest virtual water inflow, reaching 20.367 billion $m^3$, followed by Henan Province and Inner Mongolia Autonomous Region, while Qinghai Province has the smallest virtual water inflow, accounting for only about 5% of the virtual water inflow in Shandong Province. When examining output, Henan Province has the largest virtual water outflow, followed by Inner Mongolia, Shandong and Gansu Province. Shandong Province is located in the eastern coastal area, with frequent export trade. In the virtual water outflow structure, more than 60% of the virtual water flows to the international market through export trade. Overall, the inflow of virtual water in Shandong and Shanxi is far greater than the outflow, while the outflow of virtual water in Gansu Province is far greater than the inflow. In addition, the proportion of virtual water inflow and outflow of YRB provinces is higher than external provinces.

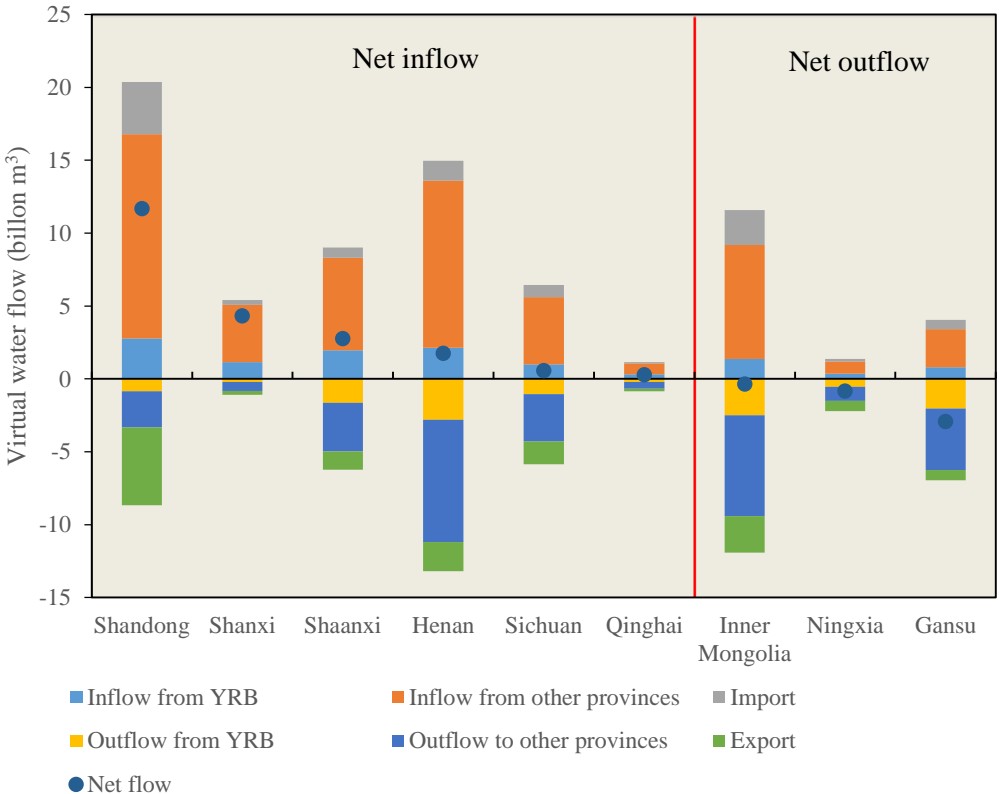

**Figure 2.** Virtual water flows and compositions of different provinces in the YRB.

Hence, in the economic development of YRB provinces, the shortage of water resources has been alleviated through virtual water trade, especially for Shandong Province and Shanxi Province. To a certain extent, this has made up for the shortage of water resources in the local development, resulting in sustainable social and economic development of water-scarce areas.

### 3.2. Interprovincial and International Virtual Water Trade of Provinces in the Basin

Although the YRB as a whole presents a favorable state of virtual water net inflow, there are significant differences in the geographical location, economic development level and resource endowment between the nine provinces in the basin. Therefore, it is necessary to further refine the research on the status of the provinces in the YRB in the virtual water trade at home and abroad. Figure 3 shows the net flow of virtual water in interprovincial and international trade. In interprovincial trade, Inner Mongolia, Gansu, and Ningxia are in a state of virtual water net outflow, and the other six provinces are in the state of virtual water net inflow. Among them, Gansu has the largest net outflow of virtual water, which is 285 million m$^3$. Shandong has the largest net inflow of virtual water, which is 1.345 billion m$^3$. Overall, most provinces in the YRB import a lot of virtual water from other provinces through interprovincial trade, which is conducive to easing the pressure of local production water consumption. In international trade, except Shanxi Province, all other provinces are in the state of virtual water net outflow. Shandong has the largest net export volume of virtual water, reaching 1.746 billion m$^3$, followed by Sichuan and Henan. Water resources in the YRB are in shortage overall, and water resources carrying capacity of each province is overloaded, while international trade further aggravates the water resources shortage crisis of the YRB.

When combing the interprovincial trade and international trade, as shown in Figure 3, the nine provinces in the basin can be divided into three parts. The first part has a net-inflow due to virtual water net-inflow in interprovincial trade and international trade. For example, in Shanxi Province, large amount of virtual water net inflow reduces agricultural and industrial consumption, hence rare water resources can serve domestic consumption. The second part has a net-outflow due to virtual

water net-outflow in both interprovincial trade and international trade, including Gansu, Ningxia, and Inner Mongolia, the virtual water net-outflow further aggravates the crisis of local water shortage, which will affect the sustainable development of social economy in the region. The third part net-exports virtual water in international trade, while net-imports virtual water through interprovincial trade, such as Qinghai, Sichuan, Henan, Shaanxi, and Shandong. In general, these five provinces have net imports of virtual water. While they have net exports of virtual water in the international market. Their net-imports of more virtual water in interprovincial trade compensated for their net-export of virtual water internationally. Hence, we should consider compensation of water resources in virtual water production areas.

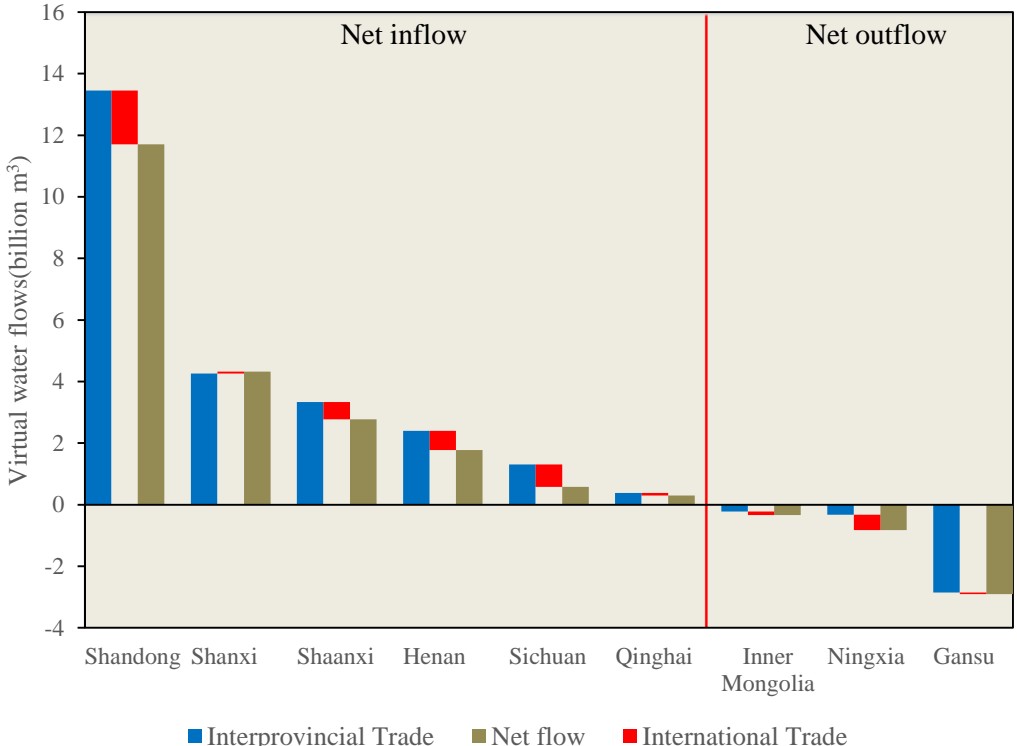

**Figure 3.** The net flow of virtual water in interprovincial and international trade.

### 3.3. Virtual Water Trade Model among Provinces in the YRB

Overall, the YRB is in the state of virtual water net inflow, mainly relying on interprovincial trade, which includes the trade between the basin and other provinces and the trade between provinces in the basin. It is of great significance to refine the trade among provinces in the basin, to reveal the role of each province in the entire water resources system of the YRB, and to determine the key path of virtual water flow.

Figure 4 shows details of the virtual water net-flow in interprovincial virtual water trade of YRB provinces. According to the net flow of virtual water, nine provinces in the basin can be divided into four categories. The first category is Shandong, Shanxi, Qinghai, and Shaanxi, each province presents the state of virtual water net-inflow, not only from external provinces but also from other YRB provinces. The second category is Henan and Sichuan, each province presents the virtual water net inflow state. Although they export virtual water in trade among nine provinces in the basin, they import virtual water from external provinces. The third category includes only Inner Mongolia, which has virtual water net-outflow. Although it imports virtual water from external provinces, it has exported more virtual water in trade with other YRB provinces, resulting in virtual water net-outflow. The fourth category is Ningxia and Gansu, which has virtual water net-outflow. They not only export

virtual water in their trade with other provinces, but also export virtual water in the trade between YRB provinces.

Through the qualitative analysis of the virtual water net flow in each province, the role of each province in the virtual water trade in the YRB is clarified. Next, the flow of virtual water among nine provinces in the YRB is analyzed quantitively to further determine the importance of each province in the water resource system of the basin.

Figure 5 shows the virtual flow between provinces in the YRB. The arc length of the outermost circle represents the sum of the inflow and outflow of virtual water in each province. Strings of the same color as the arc represent the virtual outflow of water from each province. Take the virtual flow from Shaanxi to Shandong as an example. Shaanxi's arc is dark orange, while Shandong's is red. The dark orange strings connecting Shaanxi and Shandong indicate that virtual water flows from Shaanxi to Shandong. The width of the string represents the virtual water flow amout, which is 603.7 million $m^3$. The red arc line between the outer circle of Shaanxi and the same color string indicates that the virtual water ownership has changed from Shaanxi to Shandong.

From the arc length of the outer circle, we can see the virtual water trade volume of each province. Trade volume of Henan, Inner Mongolia, Shandong, Shaanxi, and Gansu is relatively large, while the virtual water trade volume of Qinghai and Ningxia is relatively small. At the same time, it can be seen from the diagram that Inner Mongolia, Henan, and Gansu are the main provinces of virtual water outflow, accounting for 62% of the total virtual water outflow, providing relatively sufficient water resources for the development of other provinces in the basin, and playing the role of virtual water supplier. In Shandong, Henan, and Shaanxi, the main provinces receiving virtual water inflow accounted for 58% of the total virtual water inflow, playing the role of virtual water consumers.

Economic ties of the provinces in the middle and lower reaches of the Yellow River are relatively close. From Table 3, the top 10 trade exchanges of virtual water flow are between the provinces in the middle and lower reaches of the Yellow River, except for the virtual water from Gansu to Shandong. The largest virtual flow is from Henan Province to Shandong Province, with a flow of 761 million $m^3$. While economic ties between the middle and upper reaches of the Yellow River are relatively weak, and the 10 trade exchanges with the smallest virtual flow are all between the middle and upper reaches of the Yellow River except for the virtual water from Shandong to Qinghai.

**Table 3.** Comparison between the largest 10 and the smallest 10 areas of virtual flow.

| The Largest 10 | | The Smallest 10 | |
|---|---|---|---|
| Henan to Shandong | 7.61 | Qinghai to Shaanxi | 0.22 |
| Henan to Shaanxi | 7.22 | Qinghai to Sichuan | 0.22 |
| Inner Mongolia to Henan | 6.70 | Shanxi to Sichuan | 0.13 |
| Gansu to Shandong | 6.41 | Shandong to Qinghai | 0.12 |
| Shaanxi to Shandong | 6.00 | Shanxi to Gansu | 0.10 |
| Inner Mongolia to Shandong | 5.01 | Shanxi to Ningxia | 0.09 |
| Inner Mongolia to Shaanxi | 4.19 | Ningxia to Qinghai | 0.09 |
| Shaanxi to Henan | 4.19 | Qinghai to Shanxi | 0.06 |
| Henan to Inner Mongolia | 3.98 | Qinghai to Ningxia | 0.03 |
| Inner Mongolia to Shanxi | 3.84 | Shanxi to Qinghai | 0.03 |

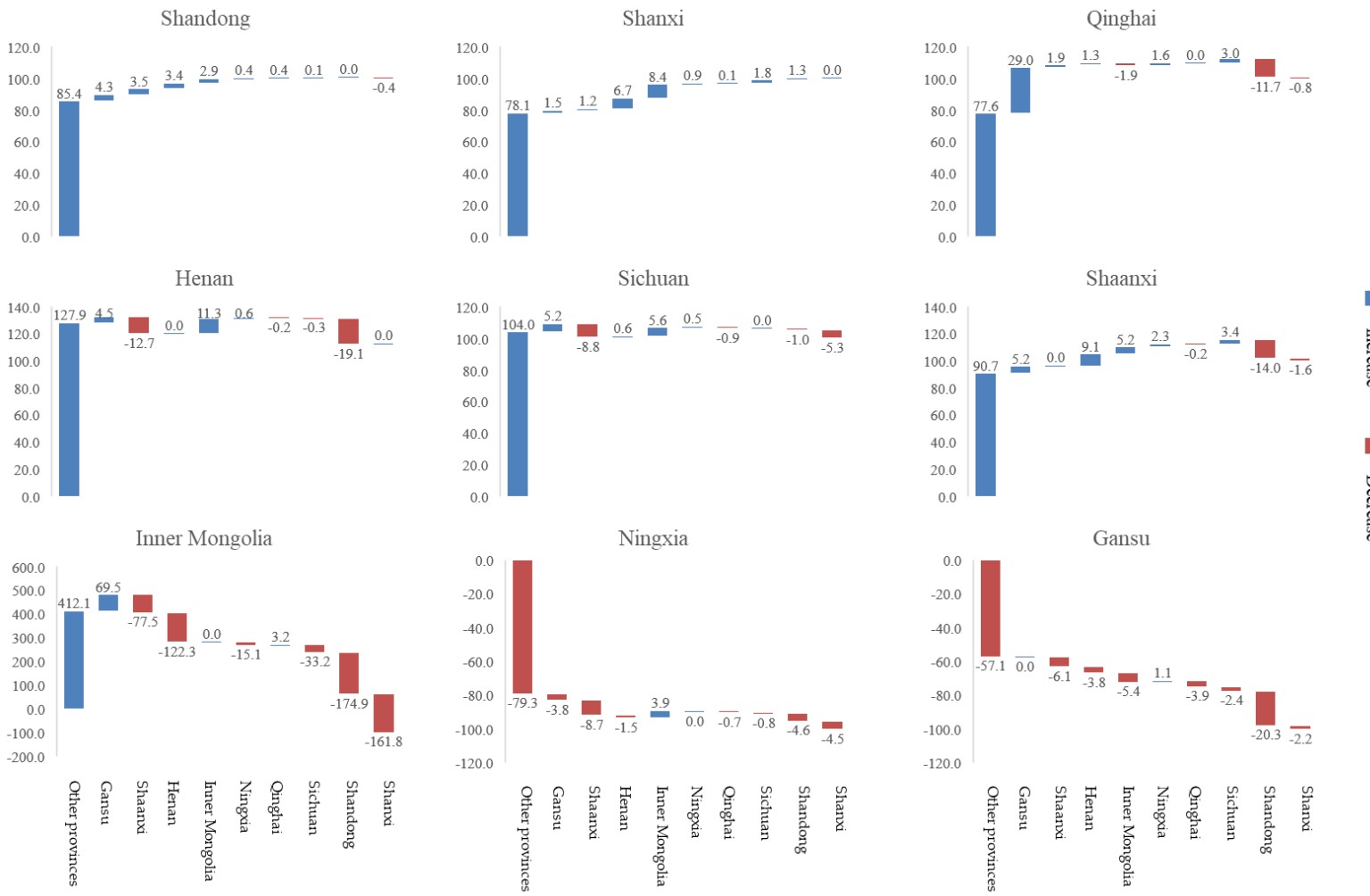

**Figure 4.** Virtual water net flow structure of each province in the YRB.

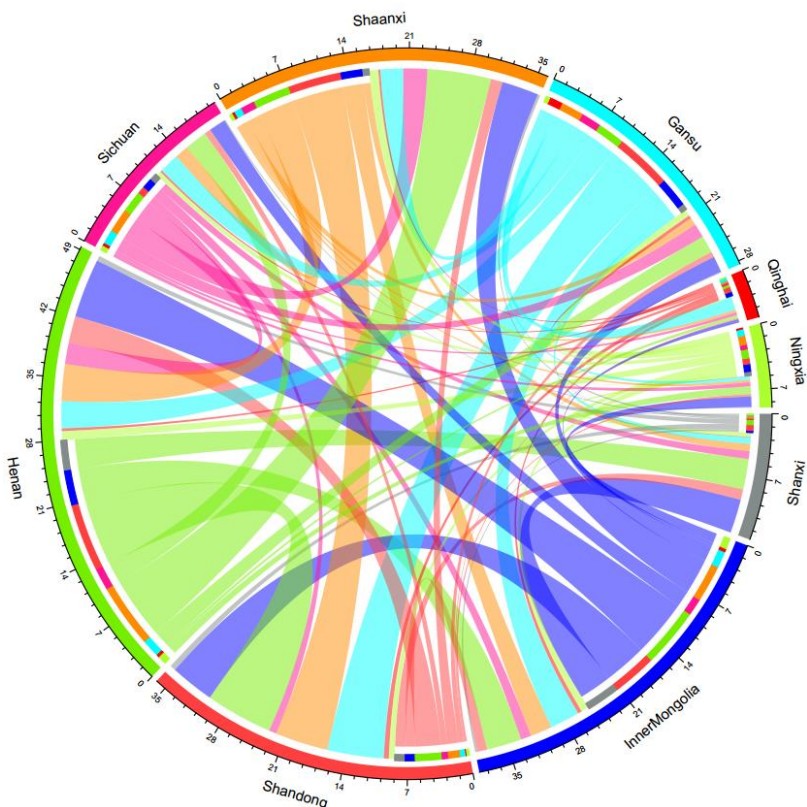

**Figure 5.** The virtual flow between provinces in the YRB.

Shandong, Shanxi, Qinghai, and Shaanxi have alleviated the local production water consumption through import of virtual water from external provinces, and further improved the situation of water shortage through import virtual water from trade within the basin. In particular, Shandong and Shanxi are not only in the state of virtual water net inflow but also in large amount. Although Henan, Inner Mongolia and Sichuan import virtual water from external provinces, they also export virtual water to other YRB provinces Henan and Sichuan maintain the status of virtual water net-inflow, while Inner Mongolia exports large amount of virtual water in the basin, resulting in the status of virtual water net-outflow. Ningxia and Gansu are in the status of virtual water net-outflow in trade with external provinces, and further export virtual water within the basin by large amount.

### 3.4. Virtual Flow Structure of Sectors of each Province in the YRB

Based on the analysis of interprovincial trade in the YRB, the role of each province in virtual water trade is determined. In order to further understand the industry structure differences in virtual water trade in each province, it is necessary to understand impacts the virtual water flow situation on different economic sectors of each province. This is helpful for the government to identify the key sectors in the virtual water trade, and provide the theoretical basis for formulating reasonable industrial structure adjustment measures and to optimize water resources allocation.

In order to capture the industrial characteristics of nine provinces in the YRB, 30 small sectors in the input-output table are combined into 7 large sectors (Table 4) according to their industrial attributes to highlight the industrial characteristics of virtual water flow. Figure 6 shows the inflow and outflow structure of the virtual water sectors in each province. In the inflow structure, the proportion of agricultural sectors in each province is the largest, followed by the manufacturing and construction industries. In the outflow structure, the proportion of agriculture in the other eight provinces is the largest, except Shanxi. Sichuan, Ningxia, and Henan also account for a large proportion of the manufacturing industry. The two largest sectors in Shanxi are mining and manufacturing. Both in the

inflow structure or outflow structure, the proportion of public utility production/supply industry and transportation industry in each province is relatively small.

**Table 4.** Thirty industrial sectors merged into seven industries.

| No. | Industry | Sector |
|---|---|---|
| 1 | Agriculture | Agriculture (including agriculture, forestry, animal husbandry and fishery) |
| 2 | Mining | Coal mining; Petroleum and gas; Metal mining; Nonmetal mining |
| 3 | Manufacturing | Food processing and tobaccos; Textile; Clothing, leather, fur, etc.; Wood processing and furnishing; Paper making, printing, stationery, etc.; Petroleum refining, coking, etc.; Chemical industry; Nonmetal products; Metallurgy; Metal products; General and specialist machinery; Transport equipment; Electrical equipment; Electronic equipment; Instrument and meter; Other manufacturing |
| 4 | Public utility production and supply | Electricity and hot water production and supply; Gas and water production and supply |
| 5 | Construction | Construction |
| 6 | Transportation | Transport and storage |
| 7 | Services | Wholesale and retailing; Hotel and restaurant; Leasing and commercial services; Scientific research; Other services |

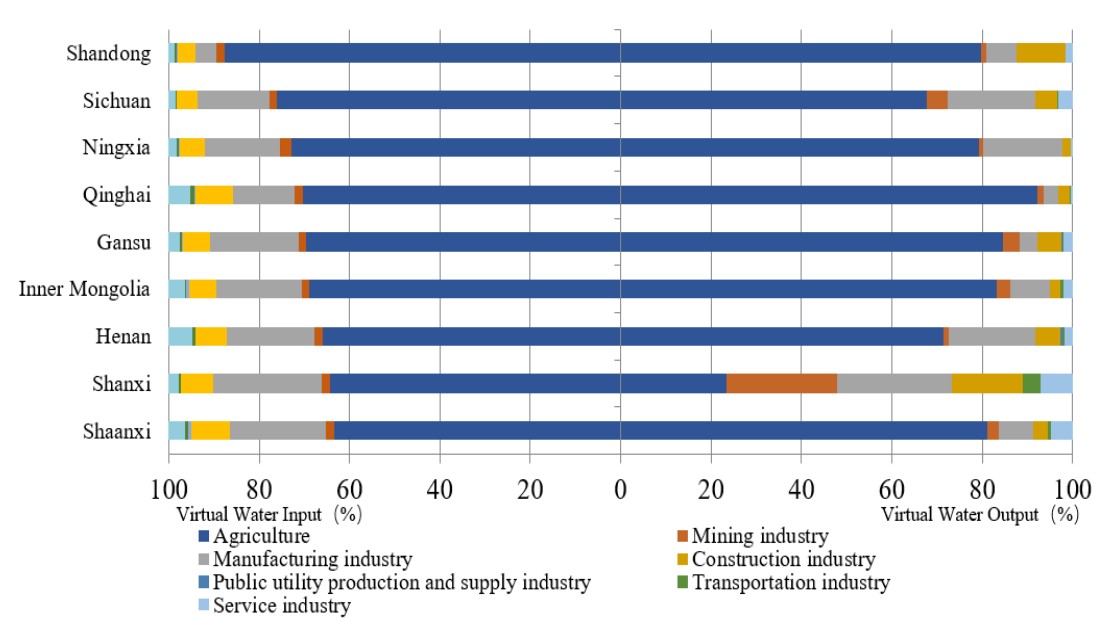

**Figure 6.** Input and output structure of sectors in each province.

In order to further study the virtual water outflow characteristics of each province in the industrial sector, we present the virtual water net flow of each province in Figure 7. In the agricultural sector, Shanxi, Shandong, and Sichuan import virtual water, while the other six provinces export virtual water. Shanxi, Gansu, and Inner Mongolia export virtual water in mining industry, and in other provinces, the net flow of virtual water in mining industry is relatively small respective to other industries. The manufacturing sector in Gansu, Shaanxi, Inner Mongolia, and Qinghai have net inflow of virtual water, while the manufacturing sector in Shandong, Sichuan, and Ningxia have net outflow of virtual water. The construction sector in Shanxi and Shandong have net outflow of virtual water, while those in Shaanxi, Qinghai, Ningxia, and Inner Mongolia have net inflow of virtual water. The proportion of virtual water net flow in public utilities production/supply industry and transportation industry in each province is relatively small and relatively balanced.

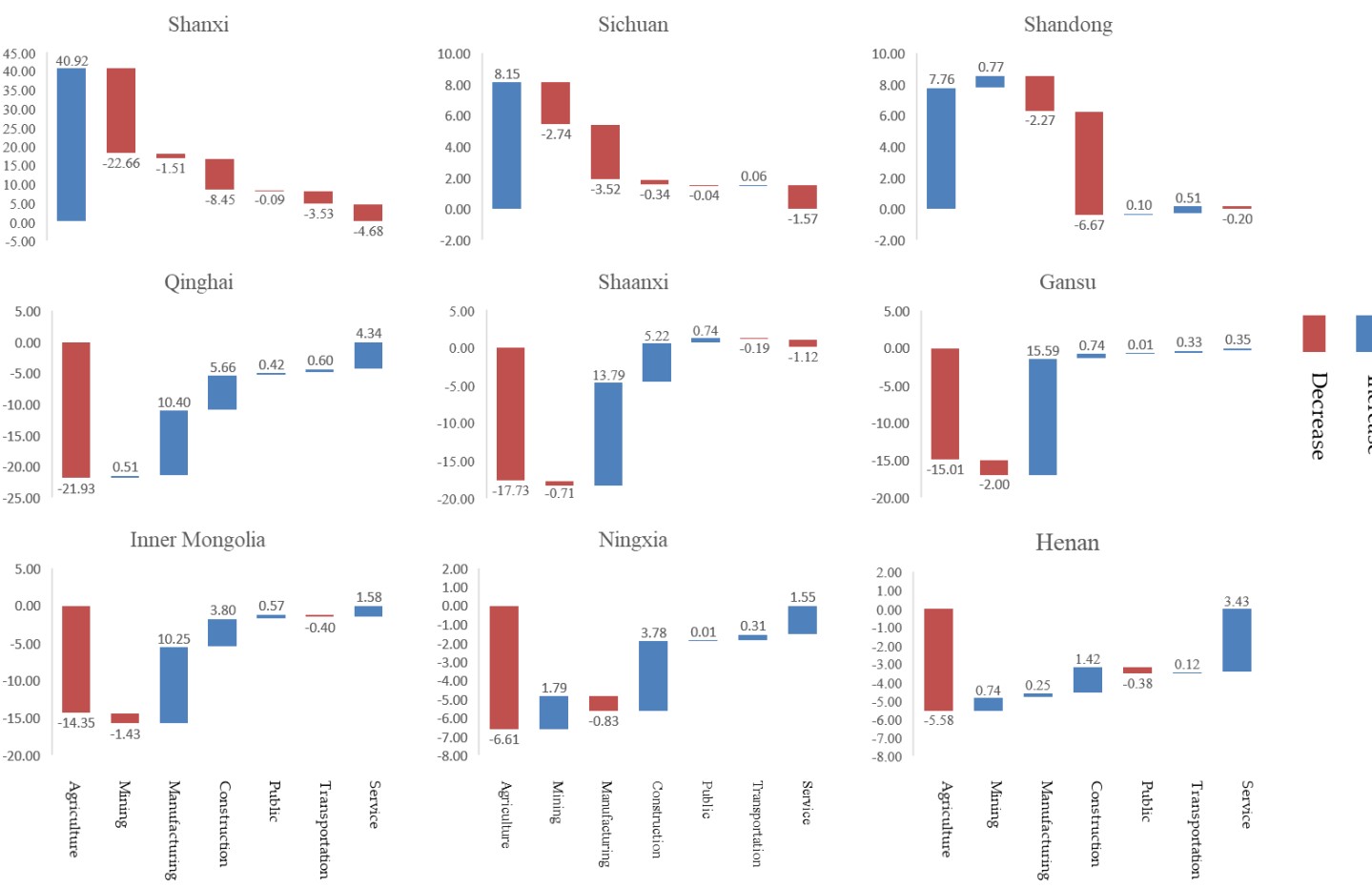

**Figure 7.** Virtual water flow structure of each sector of nine provinces in the YRB.

Through the above analysis, agriculture is the key sector to achieve water saving in each province, and most of them are in the state of virtual water net outflow. Therefore, it is particularly important to improve the efficiency of agricultural irrigation and develop modern agriculture. The main sector of net outflow of virtual water in relatively developed provinces is the industrial sector, while the agricultural sector is in the state of net inflow. On the contrary, agriculture is the key sector of virtual water output for the economically developing provinces. Therefore, according to the development stage of each province, planning the overall development of the basin, according to the comparative advantage industries of each province, and through the way of virtual water trade, adapting water resource management methods will help to solve the mismatch between regional resource endowment and economic development need.

## 4. Conclusions

### 4.1. The YRB is in the State of Net Input Virtual Water

In 2012, 74.358 billion $m^3$ of virtual water was imported into YRB, and 56.971 billion $m^3$ of virtual water was exported, hence the net inflow of virtual water is 17.387 billion $m^3$. Although the YRB plays a role as a net exporter in international virtual water trade, the virtual water inflow of interprovincial trade is much larger than the outflow of international trade. Therefore, overall, the YRB relies on interprovincial trade to import a large number of virtual water, which alleviates the shortage of water resources in the regional production and development.

### 4.2. Different Virtual Flow Paths in Nine YRB Provinces

Although the YRB overall is in the state of virtual water net inflow, the virtual flow paths of each province are different (Table 5). Shanxi Province overall presents the state of virtual water net inflow, and it is in a favorable position of virtual water net inflow in interprovincial trade and international trade. Shandong, Shaanxi, and Qinghai are in the state of virtual water net inflow overall. They net export virtual water through international trade, but they import more virtual water through interprovincial trade. Henan and Sichuan export virtual water in international trade, and further export virtual water in the basin, but by introducing more virtual water from other provinces, they ensure a favorable situation of net inflow in the virtual water trade. Inner Mongolia net exports virtual water in international trade, and imports virtual water through trade with other provinces, but exports more virtual water to the basin, resulting in virtual water net outflow overall. Ningxia and Gansu are both in the position of virtual water net outflow in international trade and interprovincial trade.

**Table 5.** Virtual water net flow path of each province in the YRB.

| Province | International Trade | Interprovincial Trade | | Total |
|---|---|---|---|---|
| | | with Other Provinces | within the Basin | |
| Shanxi | Net inflow | Net inflow | Net inflow | Net inflow |
| Shandong | Net outflow | Net inflow | Net inflow | Net inflow |
| Shaanxi | Net outflow | Net inflow | Net inflow | Net inflow |
| Qinghai | Net outflow | Net inflow | Net inflow | Net inflow |
| Henan | Net outflow | Net inflow | Net outflow | Net inflow |
| Sichuan | Net outflow | Net inflow | Net outflow | Net inflow |
| Inner Mongolia | Net outflow | Net inflow | Net outflow | Net outflow |
| Ningxia | Net outflow | Net outflow | Net outflow | Net outflow |
| Gansu | Net outflow | Net outflow | Net outflow | Net outflow |

There exists a shortage of water resources in YRB provinces except Qinghai and Sichuan. However, Shanxi, Shandong, Shaanxi, and Henan have alleviated the shortage of water resources by introducing virtual water from other provinces. Although Inner Mongolia has a net outflow state, virtual water exported has been imported by provinces in the basin, contributing to water shortage alleviation in the

YRB. However, Ningxia and Gansu have aggravated their own water shortage due to the net outflow of virtual water.

*4.3. Different Industrial Structures in Different Provinces*

The developed provinces mainly export virtual water from the secondary industry, while the backward provinces mainly export virtual water from agriculture. For example, the main sectors of virtual water net inflow in Shandong, Shanxi, and Sichuan are all agriculture, while the main sectors of virtual water net outflow are all the secondary industry related sectors, Shandong's manufacturing and construction industry, Shanxi's mining and construction industry, and Sichuan's mining and manufacturing industry. The main sector of virtual water net outflow in Qinghai, Gansu, Ningxia, and other provinces is agriculture, while the main sectors of virtual water net outflow are manufacturing and construction.

The net flow of virtual water can reflect the industrial characteristics of each province. In Shaanxi, Inner Mongolia, Henan, Qinghai, and Gansu, located in the middle and upper reaches of the Yellow River, the agricultural sector is the pillar industry, while the development of manufacturing, construction, and other industrial sectors is relatively backward. Shandong and Shanxi, which are located in the middle and lower reaches of the Yellow River, are in the opposite situation, with manufacturing, construction, and other industrial sectors as their pillar industries.

Since agriculture is the pillar industry of most provinces in the YRB, the proportion of agriculture in the nine provinces is relatively high, so the water consumption of agricultural irrigation is relatively large, usually accounting for 60% or more of the total water consumption. In this way, water for other industries, such as manufacturing and services, will be squeezed, so the opportunity cost of agricultural production water is relatively high. Due to the shortage of water resources in the YRB, the scarcity cost (also called opportunity cost) cannot be ignored, and it is necessary to vigorously promote water-saving irrigation in the YRB, especially in Gansu Province, which is extremely short of water resources. To develop modern agriculture, according to the natural endowment of each province, we should launch characteristic agricultural products, actively expand the market of special products, and achieve the goal of economic development while saving water, so as to promote the sustainable economic growth of these areas. For Shandong and Shanxi provinces, which are relatively developed in industrial development, only by optimizing the industrial structure, improving the efficiency of water use, and increasing the value of unit water consumption can we better achieve the goal of water saving. Especially in Shanxi Province, because a large number of water resources are used in the mining industry, it is necessary to carry out detailed planning of water use indicators for the mining industry, and put an end to the waste of non-productive water.

## 5. Discussion

Through virtual water trade, some provinces in the YRB have indirectly introduced water resources to alleviate the pressure of local water shortage. However some provinces have aggravated the local water shortage in trade. As shown in Figure 8, Gansu, Ningxia, and Inner Mongolia net export virtual water through virtual water trade, especially in areas such as Gansu and Ningxia, where water resources are extremely scarce, the indirect export of water resources further increases the burden of local water supply. As the economic pillar of Gansu and Ningxia is agriculture, the implementation of water-saving irrigation in those areas will help to relieve the pressure of water supply, not only through the promotion of water-saving irrigation technology, but also by characteristic agriculture. Meanwhile, tourism and other third industries with high value-adding can provide diversified opportunities for economic growth.

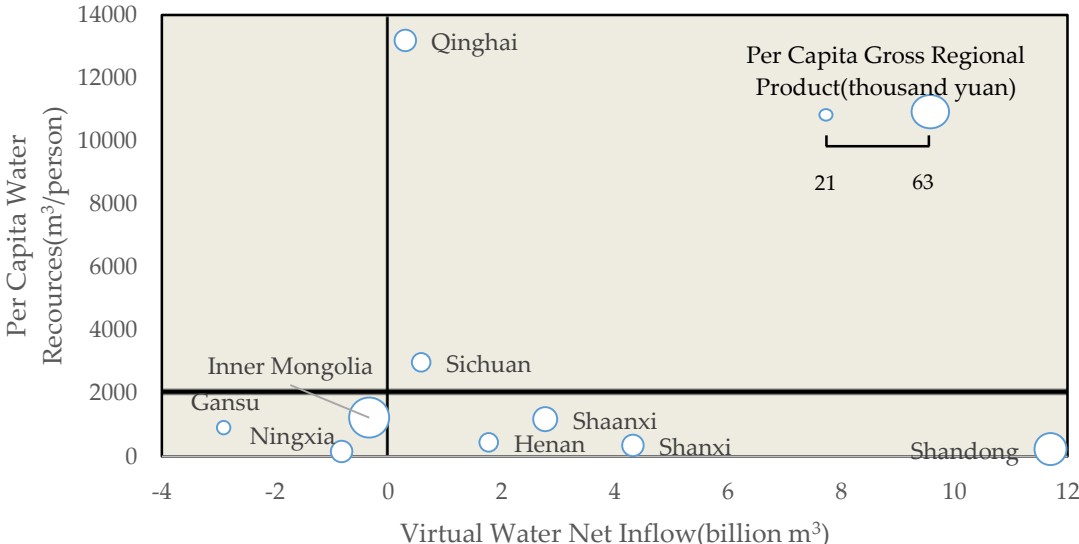

**Figure 8.** The relationship between virtual water flow and natural resource endowment and economic development level.

The net inflow of virtual water has a positive correlation with the level of economic development. Economy of provinces located in the middle and lower reaches of the Yellow River are generally more developed, and almost all of them have virtual water net inflow. Due to the mature industrial development in the middle and lower reaches of the Yellow River, the value created by the unit water resources is greater. The agricultural products with low added value can be replaced through commodity trade to reduce the consumption of water resources. For example, the virtual water export intensity of Gansu, Qinghai, and Ningxia is relatively high (the virtual water volume implied by the 10,000 yuan product export), which are 462.98, 273.48, and 696.48 $m^3$/CNY. However, the export intensity of virtual water in Shandong and Shanxi is lower than 100 $m^3$/CNY, while the added value of the water resources that replace the agricultural products is only occupied by the provinces in the middle and lower reaches, but not allocated to the provinces providing virtual water. Therefore, those provinces that provide virtual water through agricultural products need to be compensated, especially those with severe water shortage such as Gansu. Compensation needs to be provided by the provinces that consume agricultural products [41].

In conclusion, water quality protection and water conservation are necessary to achieve high-quality, sustainable development of the regional economy on the premise of reducing ecological damage, in both the rich water area and the poor water area. Through the carrying products, virtual water trade can transfer values of water resources between regions with relative ease, which can reduce the local water consumption in the production process and therefore reduce the pressure of water supply. More importantly, virtual water trade compensates ecological impact to production related water consumption, providing ecological sustainability of production expansion. Therefore, in order to achieve the high-quality development of the YRB, the virtual water trading strategy can help to sustain the development of all provinces in the basin overall. While strengthening trade between YRB provinces, an appropriate compensation method for the virtual water net outflow area matched with responsibilities of the virtual water net inflow area should be further developed.

**Author Contributions:** G.T. carried out the concepts and definition of the study, analyzed the data and wrote the manuscript; X.H. performed the statistical analysis and designed the figures; C.Z. carried out the design of intellectual content and modified the manuscript; J.L. (Jiaojiao Li) provided assistance for data acquisition, data analysis and statistical analysis; J.L. (Jining Liu) collected important background information. All authors have read and agreed to the published version of the manuscript.

**Acknowledgments:** We would like to acknowledge the support of the National Social Science Fund Project (Grant No. 17ZDA064); National Natural Science Foundation of China (Grant No. 41471456), and Fundamental Research

Funds for the Central Universities (Grant No. 2019B19814). We also acknowledge the editors and reviewers for their valuable comments.

**Conflicts of Interest:** The authors declare no conflicts of interest.

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
