# Peer review of "Virtual Water Flows Embodied in International and Interprovincial Trade of Yellow River Basin: A Multiregional Input-Output Analysis"

_sustainability, doi:10.3390/su12031251_

Round 1
Reviewer 1 Report
At general level, it is useful to underline that that to use a lot of water where this is available in sufficient quantity to satisfy all uses, both anthropic and ecosystem, it cannot be considered a negative value.
The case of water is also complicated by the fact that it is on the one hand an asset whose availability is cyclically renewed, and on the other it has a "flowing" nature, with limited possibilities of being stored and preserved. In other words, it is necessary to consider that use is not necessarily subtracts possibility of alternative use (the same water, returned to the circulation, will be again available); and, on the other hand, that non-use does not necessarily result in savings, from when the water that flows without being used ends up at sea sooner or later.
The relevant components for the economic analysis therefore concern not so much the consumption in as such, but the concentration of consumption over time. The availability of the resource must be assessed instant by instant, and a significant impact in economic terms can occur, for example, because at that precise moment the use "x" prevents or makes the use "y" more difficult, or because it draws on a stock (for example a stratum, a lake) which takes away from someone else the future option of use the same stock.
Affirming that water is an "economic good" has nothing to do with one of its own transformation into a commodity to be exchanged for a price; but it concerns the need to record in the costs all the sacrifices that a given use entails. Sacrifices that can be roughly distinguished in three categories:
financial costs: these are the economic resources that must be used to return water is available (e.g. lifting, transport, treatment) and to remove it after use, returning it to the environment (sewage, treatment, sludge disposal, etc.): scarcity costs (sometimes referred to as "resource costs"): these are alternative economic values which are sacrificed in the event that one use prevents another.
For example, if the use by the farmer x prevents the use of the farmer y, the cost of scarcity will equal the economic value of the production of y (equal to the market value of its output, net of the production costs it should bear);
environmental costs (sometimes referred to as "negative externalities"): this is the value of the component’s ecosystems, landscapes or similar on which a specific use of the resource impacts. For example, the pollution caused by the discharge of water contaminated by use could lead to one reduction of the ecological quality of a water body, depriving it of certain environmental functions.
These costs not always occur and they are not easily associated with a specific method of use or impact. The cost of the resource is revealed not only because there is someone who uses water, but also because there is someone else who would like to use it instead; an environmental cost is difficult to correlate with quantity of water withdrawn, but instead depends on when, where and how it is drawn (e subsequently released).
The literature on water footprint and "virtual water" has certainly gained the awareness that not all uses are the same, and therefore before to determine whether a certain use involves an impression or not, it is necessary to distinguish according to the way with which it occurs: for example, distinguishing between consumption of "blue" water (surface or ground water) groundwater) and "green" (rainwater retained by the soil as humidity), or, in the case of first, depending on the renewable profiles of the resource used, or even distinguishing uses "Dissipative" and "non dissipative", where dissipation essentially consists of subtraction availability for a specific period of time.
However, while approaching the economic definition of impact, these indicators are still too general, as not necessarily the same amount of water taken from the same source in the same way impacts in the same way (for more further details, see in this same volume "Not all drops of water are the same" by M. Antonelli and F. Greco). These indicators are therefore useful for an overview and for identify the regions in which a stressful situation is most likely to be identified, but not they are sufficient in themselves to characterize it.
Reviewer 2 Report
I am sure that your submission paper is very useful for the future plans of the water resources. Also, the application site, China, is challenging for the groups of water engineers. Therefore I truly hope that some of the materials and contexts in your paper to be improved for the readers in the future.
Reviewer 3 Report
In my opinion the only issue is the low quality of figures, they have to be improved before the acceptance.
Reviewer 4 Report
Please find my comments in the attached file.
